# LSH MICROBATCHES FOR STOCHASTIC GRADIENTS: VALUE IN REARRANGEMENT

## ABSTRACT

Metric embeddings are immensely useful representations of associations between entities (images, users, search queries, words, and more). Embeddings are learned by optimizing a loss objective of the general form of a sum over example associations. Typically, the optimization uses stochastic gradient updates over minibatches of examples that are arranged independently at random. In this work, we propose principled methods of structuring the *arrangements* of training examples aimed at accelerating the training. Our arrangements consist of randomized *microbatches* of examples from a distribution that is guided by the structure of the example associations but respects a specified marginal distribution of training examples. We present efficient microbatch generators and experimentally demonstrate training time acceleration of 3-30%. Structured arrangements emerge as a powerful and novel performance knob for SGD that is independent and complementary to other SGD hyperparameters and thus is a candidate for wide deployment.

## 1 INTRODUCTION

Metric embeddings of entities that are trained to capture example associations are common representations that also allow for inference of associations not present in the data. Embeddings are used in complex learning tasks or directly applied for similarity and recommendations tasks. Example usage domains includes embeddings of text document from occurrences of words Berry et al. (1995); Dumais (1995); Deerwester et al. (1990), users and videos from watch or ratings Koren et al. (2009), words from co-occurrence frequencies in a corpus Mikolov et al. (2013), and nodes in a graph from co-occurrence in short random walks Perozzi et al. (2014). The example associations may involve entities of the same type (word co-occurrences, video co-watch, social) or different types (such as users and products) and often are distilled by reweighing frequencies of raw interactions Salton & Buckley (1988); Deerwester et al. (1990); Mikolov et al. (2013); Pennington et al. (2014).

Embeddings are computed by minimizing a loss objective of the form of a sum over example associations. The optimization starts with random initialization followed by gradient updates. In modern applications, the objective can have billions of terms or more, and the de facto method at such scale is stochastic gradient descent (SGD) Robbins & Siegmund (1971); Koren (2008); Salakhutdinov et al. (2007); Gemulla et al. (2011); Mikolov et al. (2013). The terms (examples) are randomly grouped into minibatches. Gradient updates, that are equal in expectation to the full gradient, are then computed sequentially for each minibatch. SGD is much more efficient than working with full gradients and the minibatch size determines the amount of concurrency. There are numerous tunable parameters and methods aimed to improve quality and speed of convergence. Some notable recent work includes per-parameter tuning of the learning rate Duchi et al. (2011) and altering the distribution of training examples by gradient magnitude Alain et al. (2015), negatives selection with triplet loss Schroff et al. (2015), clustering Fu & Zhang (2017) and diversity criteria Zhang et al. (2017).

In this work we introduce principled schemes that control the *arrangement* of examples into minibatches. Note that arrangement tuning is separate and orthogonal to optimizations knobs that alter the distribution of training examples, learning rate, or minibatch size. The baseline practice of *independent* arrangements places examples into minibatches independently at random. This practice is supported by classic SGD convergence analysis and has the upside of controlling the variance of the stochastic gradients. We make a novel case here for an antithesis of independent arrangements

which we term *coordinated* arrangements. Coordinated arrangements are much more likely to place corresponding associations in the same minibatch. We show that coordination offers different upsides: At the micro level, updates are more effective in pulling vectors of similar entities closer. At a macro level, the examples in small fractions of epochs encode (in expectation) the similarity structure in the full set of example associations whereas independent arrangement disperse that information. This "self similarity" of the training sequence effectively allows a single epoch to act as multiple passes.

We specify coordinated arrangements through a distribution on randomized subsets of associations which we refer to as *microbatches*. Basic coordinated microbatches co-place corresponding associations to the maximum extent possible while adhering to the marginal distribution of training examples. Locality Sensitive Hashing (LSH) maps allow for refining our microbatches so that corresponding associations are more likely to be co-placed when the overall similarity of the entities is higher. The LSH maps we apply leverage some available coarse proxy of entity similarity. A readily available first-order proxy is the similarity of the sparse association vectors. Another proxy is an embedding obtained by a weaker model. We present efficient generators of basic and refined microbatches for both LSH functions. Finally, microbatches are independently grouped into minibatches of desired size, which allows us to retain the traditional advantages of independent arrangements at the microbatch level. This design enables us to tune the microbatches to the problem and even the stage of training.

We compare the effectiveness of different arrangements through experiments on synthetic stochastic block matrices and on recommendation data sets. The stochastic block data, with its simplicity and symmetry, allows us to factor out the potential effect of other optimizations. We learn that basic coordination is always beneficial earlier in training whereas LSH refinements are more effective later on. We obtain consistent 3%-30% reduction in training with respect to the independent arrangements baseline that holds across other training hyperparameters.

The paper is organized as follows. Section 2 presents necessary background on the loss objective we use in our experiments and working with minibatches with one-sided gradient updates. In Section 3 we define LSH microbatches and coordinated minibatch arrangements. In We report our experiment results comparing different arrangement methods on stochastic blocks and on recommendation data sets in Section 4. In Section 5 we examine properties of coordinated arrangements that are helpful for faster convergence. We conclude in Section 6.

## 2 PRELIMINARIES

Our data has the form of associations between a set $F$ of *focus* entities and a set $C$ of *context* entities. In practice, the focus and context entities could be the same type or even two representations of the same set (say words or images) or different types (users and videos). We use $\kappa_{ij}$ as the association strength between focus $i$ and context $j$. In practice, the association strength can be derived from frequencies in the raw data or from an associated value (for example, numeric rating or watch time).

An embedding is a set of vectors $\boldsymbol{f}_i, \boldsymbol{c}_j \in \Re^d$ that is trained to minimize a loss objective that encourages $\boldsymbol{f}_i \cdot \boldsymbol{c}_j$ to be larger when $\kappa_{ij}$ is larger. For concreteness, we focus here on Skip Gram with Negative Sampling (SGNS) loss Mikolov et al. (2013). Examples of positive associations $(i, j)$ are drawn with probability proportional to $\kappa_{ij}$. Random associations are then used as negative examples Hu et al. (2008): Each positive example $(i, j)$ is matched with a set of negative examples of $i$ with random context entities and similarly $j$ with random focus entities. The negative examples provide an antigravity effect that prevents all embeddings from collapsing into the same vector. We use the notation $n_{ij}$ for respective weights of negative associations.

The SGNS objective is designed to maximize the log likelihood of these examples. The probability of positive and negative examples are respectively modeled using

$$p_{ij} = \sigma(\boldsymbol{f}_i \cdot \boldsymbol{c}_j) = \frac{1}{1 + \exp(-\boldsymbol{f}_i \cdot \boldsymbol{c}_j)} \ \text{ and } \ 1 - p_{ij} = \sigma(-\boldsymbol{f}_i \cdot \boldsymbol{c}_j) = \frac{1}{1 + \exp(\boldsymbol{f}_i \cdot \boldsymbol{c}_j)} \ .$$

The likelihood function, which we seek to maximize, can then be expressed as $\Pi_{ij} \, p_{ij}^{\kappa_{ij}} \, \Pi_{ij} (1 - p_{ij})^{n_{ij}}$. We equivalently can minimize the negated log likelihood that turns the objective into a sum:

$$L := \sum_{ij} \kappa_{ij} \log p_{ij} + \sum_{ij} n_{ij} \log(1 - p_{ij}) \ .$$

The optimization is performed by random initialization of the embedding vectors followed by stochastic gradient updates. The stochastic gradients are computed for minibatches of examples that include $b$ positive examples, where $(i, j)$ appears with frequency $\kappa_{ij}/|\kappa|_1$ and corresponding sets of negative examples.

## 2.1 ONE-SIDED UPDATES

We work with *one-sided* updates, where each minibatch updates only its focus or only its context embedding vectors, and accordingly say that minibatches are *designated* for focus or context updates. One-sided updates are related to alternating minimization Csiszar & Tusnády (1984) and to decomposition-coordination approaches Cohen (1980). For our purposes, one-sided updates facilitate our coordinated arrangements and also allow more precise matching of corresponding sets of negative examples to positive ones. In a focus-updating minibatch, we will generate a random set of $\lambda$ context vectors $C'$ and for each positive example $(i, j)$ we generate $\lambda$ negative examples $(i, j')$ for $j' \in C'$. The focus embedding $\boldsymbol{f}_i$ is updated to be closer to $\boldsymbol{c}_j$ but at the same time repealed (in expectation) from $C'$ context vectors. With applicable learning rate $\eta$, the update has the form:

$$\boldsymbol{f}_i \leftarrow \boldsymbol{f}_i - \eta \nabla_{\boldsymbol{f}_i} \left( \log \sigma(\boldsymbol{f}_i \cdot \boldsymbol{c}_j) + \sum_{j' \in C'} \log \sigma(-\boldsymbol{f}_i \cdot \boldsymbol{c}_{j'}) \right) .$$

Symmetrically, for a context-updating minibatch we use a random set of focus vectors $F'$ as our negative examples and for each positive example $(i, j)$ we perform the update $\boldsymbol{c}_j \leftarrow \boldsymbol{c}_j - \eta \nabla_{\boldsymbol{c}_j} \left( \log \sigma(\boldsymbol{f}_i \cdot \boldsymbol{c}_j) + \sum_{i' \in F'} \log \sigma(-\boldsymbol{f}_{i'} \cdot \boldsymbol{c}_j) \right).$

## 3 MINIBATCH ARRANGEMENT SCHEMES

Minibatch arrangement schemes determine how examples are organized into minibatches of specified size parameter $b$. At the core of each arrangement scheme is a distribution $\mathcal{B}$ over subsets of positive examples which we call *microbatches*. Our microbatch distributions have the property that the marginal probability of each example $(i, j)$ is always equal to $\kappa_{ij}/||\kappa||_1$. However, subset probabilities vary between schemes and within a scheme we generally will have different distributions $\mathcal{B}_f$ for focus and $\mathcal{B}_c$ for context designations.

Minibatches are obtained from microbatches as specified in Algorithm 1 for focus updates (a symmetric construction applies to context updates). The input is a microbatch distribution $\mathcal{B}_f$, minibatch size parameter $b$, and a parameter $\lambda$ that determines the ratio of negative to positive training examples. We draw independent microbatches until we have a total of $b$ or more positive examples. We then draw $\lambda$ random contexts $C'$ and generate $\lambda$ negative examples $(i, j')$ for $j' \in C'$ for each positive example $(i, j)$. When training, we alternate between focus and context updating minibatches.

---

**Algorithm 1:** Minibatch construction (Focus updates)

---

**Input:** $\mathcal{B}_f, b, \lambda$ `// Microbatch distribution, size, negative sampling`
$P, N \leftarrow \emptyset$
**repeat** $X \sim \mathcal{B}_f; P \leftarrow P \cup X$
  **until** $|P| \geq b$
$C' \leftarrow \lambda$ contexts selected uniformly
  at random
**foreach** *example pair* $(i, j) \in P$ **do**
  **foreach** $j' \in C'$ **do**
    $\lfloor\ N \leftarrow N \cup \{(i, j')\}$
**return** $P \cup N$

---

The baseline independent arrangement method (IND) can be placed in this framework using microbatches that consist of a single positive example $(i, j)$ selected with probability $\kappa_{ij}/||\kappa||_1$ (see Algorithm 2). With coordinated arrangements, the microbatch distribution depends on designation. Algorithm 3 generates basic coordinated microbatches (COO) for focus-updates. These microbatches have the form of a set of positive examples with a shared context.

Basic microbatches have the property that if $\kappa_{ij} \leq \kappa_{i'j}$ and the positive example $(i, j)$ is included in a basic microbatch then the microbatch would also include the positive example $(i', j)$. It is instructive to consider the special case of $\kappa$ with all positive entries being equal: Basic microbatches with focus designation have the form of some context $j$, and all $(i, j)$ with positive $\kappa_{ij}$. Our basic microbatches maximize the co-placement probability of two examples with a shared context while respecting the marginal probabilities. A symmetric construction applies to context-update microbatches that maximize the co-placement of two

examples with a shared focus. We establish that our microbatch generator respects the marginal probabilities, that is, example $(i, j)$ is included with probability $\propto \kappa_{ij}$:

**Lemma 3.1.** *A positive example $(i, j)$ is included in a basic coordinated microbatch (Algorithm 3) with focus designation with probability $\kappa_{ij}/M$ and in a microbatch with context designation with probability $\kappa_{ij}/N$, where $M := \sum_h M_h$ with $M_h := \max_i \kappa_{ih}$ for context $h$ and $N := \sum_h N_h$ with $N_h := \max_j \kappa_{hj}$ for focus $h$.*

*Proof.* Consider focus updates (apply a symmetric argument for context updates). The example $(i, j)$ is selected if first context $j$ is selected, which happens with probability $M_j/M$ and then we have $u \leq \kappa_{ij}/M_j$ for independent $u \sim U[0, 1]$, which happens with probability $\kappa_{ij}/M_j$. Combining, the probability that $(i, j)$ is selected is the product of the probabilities of these two events which is $\kappa_{ij}/M$. □

We preprocess $\kappa$ and precompute the per-context maxima so that we can efficiently draw a random context with probability proportional to the column maxima. We also generate an index for efficient retrieval, for context $j$ and a threshold value $T$, of all entries $i$ with $\kappa_{ij} \geq T$. The preprocessing is linear in the sparsity of $\kappa$. Note that it is typical in applications to preprocess the data, in particular, to obtain $\kappa$ by aggregating and reweighing the frequencies of associations in the raw data. The additional overhead imposed by our preprocessing is minimal. We can observe that given this preprocessing, the microbatch generator is very efficient: It draws a context according to the distribution (which is $O(1)$ operation) and then uses $T$ to index into the start point in the sorted "column."

## 3.1 LSH MAPS

Placement of $(i, j)$ and $(i', j)$ in the same focus updating microbatch results in pulling $\boldsymbol{f}_i$ and $\boldsymbol{f}_{i'}$ closer together (see the micro-level property highlighted in Section 5). This is helpful when the entities $i$ and $i'$ are similar in that they have a close target embeddings. Otherwise, the update is anyhow countered by other updates and the placement have undesirable effect that it increases the microbatch size and may increase variance of the stochastic gradients. In particular, since a large microbatch is processed by consecutive same-designation minibatches, it increases the effective minibatch size to microbatch size. This suggests that it would be useful to tune the quality of co-placements so as to decrease unhelpful ones while retaining as many helpful ones as we can. We do this using locality sensitive hashing (LSH) to compute randomized maps of entities to keys. Each map is represented by a vector $\mathbf{s}$ of keys for entities such that similar entities are more likely to obtain the same key. We use these maps to refine our basic microbatches by partitioning them according to keys. The refined microbatches (COO+LSH) are smaller and of higher quality, with a larger fraction of helpful co-placements (higher "precision") but also fewer helpful co-placements over all (lower "recall").

Ideally, our LSH modules would correspond to the similarity captured by the target embedding. This however creates a chicken-and-egg problem as the target embedding is not available at the start of training and is what we want to compute. Instead, we use LSH modules that are available at the start of training and are only a coarse proxy of the target similarity. The coarse embedding can come from a weak signal evident from feature vectors, a weaker (and cheaper to train) model, or from a partially-trained model. We work with two LSH modules based on *Jaccard* and on *Angular* LSH. The modules generate maps for either focus or context entities which are applied according to the microbatch designation. We will specify the map generation for focus entities, as maps for context entities can be symmetrically obtained by reversing roles.

Our Jaccard LSH module is outlined in Algorithm 4. The probability that two focus entities $i$ and $i'$ are mapped to the same key (that is, $s_i = s_{i'}$) is equal to the weighted Jaccard similarity of their association vectors $\kappa_i$ and $\kappa_{i'}$. (For context updates the map is according to the vectors $\kappa_{\cdot j}$):

**Lemma 3.2.** *Cohen et al. (2009)*

$$\Pr[s_i = s_{i'}] = \frac{\sum_j \min\{\kappa_{ij}, \kappa_{i'j}\}}{\sum_j \max\{\kappa_{ij}, \kappa_{i'j}\}} .$$

Our angular LSH module is outlined in Algorithm 5. Here we input an explicit "coarse" embedding $\tilde{\boldsymbol{f}}_i, \tilde{\boldsymbol{c}}_j$ that we expect to be lower quality proxy of our target one. (In our experiments we use angular

LSH with a lower dimension SGNS model.) Each LSH map is obtained by drawing a random vector and then mapping each entity $i$ to the sign of a projection of $\tilde{f}_i$ on the random vector. The probability that two focus entities have the same key depends on the angle between their coarse embedding vectors:

**Lemma 3.3.** *Goemans & Williamson (1995)*

$$\Pr[s_i = s_{i'}] = 1 - \frac{1}{\pi} \cos^{-1} \cos_{sim}(\tilde{f}_i, \tilde{f}_{i'}) \,,$$

*where $\cos_{sim}(\boldsymbol{v}, \boldsymbol{u}) := \frac{\boldsymbol{v} \cdot \boldsymbol{u}}{||\boldsymbol{v}||_2 ||\boldsymbol{u}||_2}$ is the cosine of the angle between the two vectors.*

We can always apply multiple LSH maps to further refine basic microbatches. Each application decreases the microbatch size and increases quality (similarity level of entities placed in the same microbatch). More precisely, with $r$ independent maps the probability that two entities are microbatched together decreases to $\Pr[s_i = s_{i'}]^r$ – thus the probability decreases faster when similarity is lower. The number of LSH maps we apply can be set statically or adaptively to obtain microbatches that are at most a certain size (usually the minibatch size). We can also increase the number as training progresses. For efficiency, we precompute a small number of LSH maps in the preprocessing step and randomly draw from that set. The computation of each map is linear in the sparsity of $\kappa$.

---

**Algorithm 2:** Independent microbatch

**Input:** $\kappa$
Choose $(i, j)$ with probability $\kappa_{ij}/||\kappa||_1$;
**return** $\{(i, j)\}$

---

**Algorithm 3:** Basic coordinated microbatches (Focus updates)

**Input:** $\kappa$
```
// Preprocessing:
```
**foreach** *context $j$* **do**
    $M_j \leftarrow \max_i \kappa_{ij}$ `// Maximum entry`
    `   for context j`
    Index column $j$ so that we can return for each
    $t \in (0, 1]$, $P(j, t) := \{i \mid \kappa_{ij} \geq tM_j\}$.
```
// Microbatch draw:
```
Choose a context $j$ with probability $\frac{M_j}{\sum_h M_h}$
Draw $u \sim U[0, 1]$
**return** $P(j, u)$

---

**Algorithm 4:** Jaccard LSH (Focus updates)

**foreach** *context $j$* **do** `// i.i.d Exp`
    `distributed`
    Draw $u_j \sim \text{Exp}[1]$
**foreach** *focus $i$* **do** `// assign LSH bucket`
    `key`
    $s_i \leftarrow \arg\min_j u_j/\kappa_{ij}$
**return s**

---

**Algorithm 5:** Angular LSH (Focus updates)

**Input:** $\{\tilde{f}_i\}$ `// coarse d dimensional`
    `embedding`
Draw $r \sim S_d$ `// Random vector from`
    `the unit sphere`
**foreach** *focus $i$* **do** `// assign LSH bucket`
    `key`
    $s_i \leftarrow \mathbf{sign}(r \cdot \tilde{f}_i)$
**return s**

---

## 4 ARRANGEMENT METHODS EXPERIMENTS

We train embeddings with different minibatch arrangement methods: The baseline independent arrangements (IND) as in Algorithm 2, coordinated arrangements with basic microbatches (COO) as in Algorithm 3, and the following coordinated arrangements with LSH partitioned microbatches (COO+LSH): (i) *Jaccard*: single Jaccard LSH map, (ii) *Jaccard\**: repeated partitions until the microbatch sizes are capped by the minibatch size $b$ and (iii) *angular\**: angular LSH with respect to a pre-computed $d = 3$ dimensional embedding, applied repeatedly to obtain microbatches of size that is capped by $b$. We also evaluate tunable arrangements (MIX) that start with COO, may switch to (one variant) of COO+LSH or to IND, and may switch from COO+LSH to IND. This MIX design allows us to benefit from a higher recall (albeit lower precision) of helpful co-placements (COO) being more effective in early training regime and also allows for IND arrangements in the late training regime after the coarse similarity proxy used in our LSH maps exceeds its utility. The (at most) two switch points of each MIX method are hyperparameters that were determined once via a grid search and then used across repetitions (generated synthetic data and splits for recommendation data). The coordinated arrangement methods we implemented are not meant to be a comprehensive coverage of our approach and were not even selected to maximize performance. Our intention is to present and evaluate a sampler of basic simple methods in order to provide usage examples, understand the

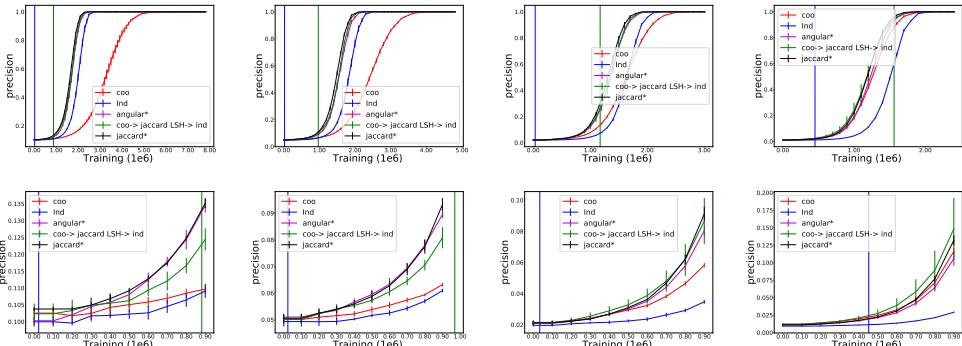

Figure 1: Precision at $k = 10$ with different arrangement methods in the course of training ($d = 50$, $b = 64$). Using $10^4 \times 10^4$ stochastic blocks matrices with $B \in \{10, 20, 50, 100\}$. The switch point for the MIX method are shown in blue (to COO+LSH) and green (to IND). The solid lines are for Jaccard LSH and the dashed lines are for angular LSH.

potential of coordinated arrangements, and develop an understanding of what methods are more effective in different training regimes.

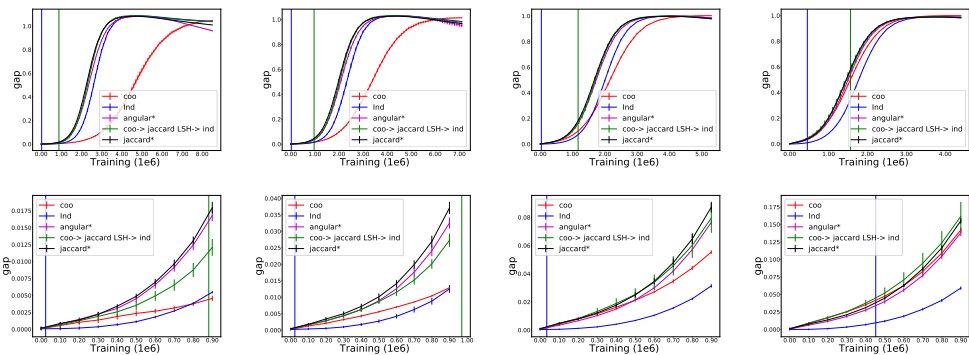

Figure 2: Cosine gap with different arrangement methods in the course of training ($d = 50$, $b = 64$). Using $10^4 \times 10^4$ stochastic blocks matrices with $B \in \{10, 20, 50, 100\}$. The switch point for the MIX method are shown in blue (to COO+LSH) and green (to IND). The solid lines are for Jaccard LSH and the dashed lines are for angular* LSH.

### 4.1 STOCHASTIC BLOCKS DATA

We generated synthetic data sets using the stochastic blocks model Condon & Karp (2001). This allowed us to explore the effectiveness of different arrangement methods when we vary the number and size of blocks. The simplicity and symmetry of this data (parameters, entities, and associations) allowed us to compare different arrangement methods while factoring out other potential optimizations and methods geared for asymmetric data such as per-parameter learning rates or altering the distribution of examples.

The parameters for the generative model are the dimensions $n \times n$ of the matrix, the number of (equal size) blocks $B$, the number of interactions $r$, and the in-block probability $p$. The rows and columns are partitioned to consecutive groups of $n/B$, where the $i$th part of rows and $i$th part of columns are considered to belong to the same block. We generate the matrix by initializing the associations to be $\kappa_{ij} = 0$. We then draw $r$ interactions independently as follows. We select a row index $i \in [n]$ uniformly at random. With probability $p$, we select (uniformly at random) a column $j \in [m]$ that is in the same block as $i$. Otherwise (with probability $1 - p$) we select a uniform column $j \in [n]$ that is outside the block of $i$. We then increment $\kappa_{ij}$. The final association $\kappa_{ij}$ is the number of

times the interaction $(i, j)$ was drawn. In our experiments we set $n = 10^4$, $r = 10^7$, $p = 0.7$ and $B \in \{10, 20, 50, 100\}$.

## 4.2 IMPLEMENTATION AND METHODOLOGY

Our implementation was in Python using the TensorFlow library Abadi & et al. (2015) and building on the word embedding implementation of Mikolov et al. (2013)[1] except that we used our methods to specify minibatches. The implementation included a default bias parameter and we trained embeddings with and without the bias parameter. The relative performance of arrangement methods was the same but the overall performance was significantly better with the bias parameter. We therefore report results of embeddings with bias parameters (updated in a two-sided manner). We used fixed and polynomially-decaying uniform learning rate with parameters that were tuned to perform well on the baseline independent arrangements and used with all arrangement methods. We worked with minibatch size parameter values $b \in \{4, 64, 246\}$ (recall that $b$ is the number of positive examples and $\lambda = 10$ negative examples are matched with each positive example), and embeddings dimension $d \in \{3, 5, 10, 25, 50, 100\}$.

## 4.3 QUALITY MEASURES

We use two measures of the quality of an embedding with respect to the blocks ground truth. The first is the *cosine gap* which measures average quality and is defined as the difference in the average cosine similarity between positive examples and negative examples. We generate a sampled set $T_+$ of same-block pairs $(i, j)$ as positive test examples and a sampled set $T_-$ of pairs that are not in the same block as negative test examples and compute

$$\frac{1}{|T_+|} \sum_{(i,j) \in T_+} \cos_{\text{sim}}(\boldsymbol{f}_i, \boldsymbol{c}_j) - \frac{1}{|T_-|} \sum_{(i,j) \in T_-} \cos_{\text{sim}}(\boldsymbol{f}_i, \boldsymbol{c}_j) . \tag{1}$$

We expect a good embedding to have high cosine similarity for same-block pairs and low (around 0) cosine similarity for out of block pairs. The second measure we use, *precision at $k$*, is focused on the quality of the top predictions and is appropriate for recommendation tasks. For each sampled representative entity we compute the entities with top $k$ cosine similarity and consider the average fraction of that set that are in the same block.

## 4.4 STOCHASTIC BLOCKS RESULTS

Representative results for $d = 50$, $b = 64$, and varying block sizes ($B = 10, 20, 50, 100$) are reported in Figure 1 (precision quality measure) and in Figure 2 (the cosine gap quality measure). For each configuration we show how quality increases in the course of training and also zoom on the early part of training. The $x$-axis in these plots shows the *amount of training* in terms of the total multiplicity of positive training examples that were used for gradient updates. Respective results for other minibatch sizes ($b = 4$ and $b = 256$) are reported in Appendix A.

We can observe that across all block sizes $B$ and for the two quality measures our coordinated arrangement methods result in faster convergence than the baseline IND method.

The different COO+LSH methods trade off the microbatch size (recall of helpful co-placements) and co-placement quality (precision). We observe that the sweet spot for this tradeoff varies for different regimes in the training. In particular, higher recall is beneficial early on: When zooming on early training we see that COO (that has the largest recall) is dominant in the very early regime but may deteriorate later, in particular for larger blocks ($B = 10$) (that yield even larger basic microbatches).

Jaccard COO+LSH that uses a single map still generates large microbatches in particular with larger blocks but when used as part of a MIX (start with COO and ends with IND) the overall training is faster that any of the components alone. We can also see that the switch point for IND occurs in early-mid training and that the training gain obtained before that point is retained.

The Jaccard* and angular* COO+LSH methods which cap the microbatch size by the minibatch size perform well for $b = 64$ and for $b = 256$. For $b = 4$, they are outperformed in early training by COO

---

[1]`https://github.com/tensorflow/models/blob/master/tutorials/embedding/word2vec.py`

(which provides a higher "recall" of helpful co-placements) and overall it is outperformed by MIX that start with COO continues with COO+LSH with a single Jaccard map, and ends with IND. The additional results reported in Appendix A for minibatch size parameters $b = 4$ and $b = 256$ indicate that (as expected) the relative advantage of the coordinated methods increases with minibatch size and consistently result int training gains of 5-30% over the baseline IND arrangements.

In Appendix B we consider training embeddings with different dimensions: We observe the same relative performance of arrangement methods as reported for $d = 50$. We also see that the peak quality is lower for $d = 3$, which justifies its use as a "coarse" proxy with angular LSH.

### 4.5 RECOMMENDATION DATA SETS AND RESULTS

We performed experiments on two recommendation data sets, MOVIELENS1M and AMAZON. The MOVIELENS1M dataset Movielen1M contains $10^6$ reviews by $6 \times 10^3$ users of $4 \times 10^3$ movies. The AMAZON dataset SNAP contains $5 \times 10^5$ fine food reviews of $2.5 \times 10^5$ users on $7.5 \times 10^3$ food items. Provided review scores were $[1\text{-}5]$ and we preprocessed the matrix by taking $\kappa_{ij}$ to be 1 for review score that is at least 3 and 0 otherwise. We then reweighed entries in the MOVIELENS1M dataset by dividing the value by the sum of its row and column to the power of $0.75$. This is standard processing that retains only positive ratings and reweighs to prevent domination of frequent entities.

| LSH | $0.75\times$ peak | | $0.95\times$ peak | | $0.99\times$ peak | |
|---|---|---|---|---|---|---|
| | %gain | $\times 10^6$ | %gain | $\times 10^6$ | %gain | $\times 10^6$ |
| AMAZON: Gain of COO+LSH over IND (peak=0.33) | | | | | | |
| Jac | 4.29 | 3.50 | 6.86 | 5.83 | 11.02 | 7.17 |
| Ang | 10.00 | 3.50 | 13.38 | 5.83 | 16.04 | 7.17 |
| MOVIELENS1M: Gain of MIX over IND (peak=0.40) | | | | | | |
| Jac | 2.13 | 1.41 | 0.58 | 1.73 | 1.55 | 1.93 |
| Ang | 4.96 | 1.41 | 8.67 | 1.73 | 11.92 | 1.93 |

Table 1: AMAZON and MOVIELENS1M: Training gain over IND baseline ($b = 64$, cosine gap).

We created a test set $T_+$ of positive examples by sampling 20% of the non zero entries with probabilities proportional to $\kappa_{ij}$. The remaining examples were used for training. As negative test examples $T_-$ we used random zero entries. We measured quality using the cosine gap equation 1 and show results averaged over 5 random splits of the data to test and training sets and 5 runs per split. The MIX and COO+LSH Jaccardwere the respective best performers on MOVIELENS1M and AMAZON. Training gains ($d = 50$) with respect to the IND baseline are reported in Table 1. We observe consistent reduction in training which indicate that arrangement tuning is an effective tool also on these more complex real-life data sets.

## 5 EXPLAINING THE UPSIDE OF COORDINATION

We highlight two properties of coordinated arrangements that are beneficial to accelerating convergence: A micro-level property that makes gradient updates more effective by moving embedding vectors of similar entities closer and a macro-level property of preserving expected similarity in fractions of epochs.

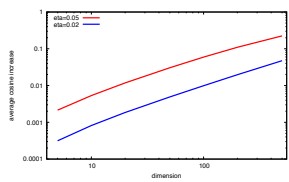

Figure 3: Expected increase in $\cos_{\text{sim}}(\boldsymbol{f}_1, \boldsymbol{f}_2)$ for $\boldsymbol{f}_i \sim \mathcal{N}^d$ after gradient update to same random context $\boldsymbol{c} \sim \mathcal{N}^d$

**Effectiveness of gradient updates**  When updates on corresponding associations of two entities are processed in the same minibatch then the cosine similarity of their embedding vectors increases. This holds also in early training when the embedding vectors are randomly initialized. Similar entities have more corresponding associations (fraction equals the Jaccard similarity) and benefit more from this property. In particular, the SGNS loss term for a positive example is $L_+(\boldsymbol{f}, \boldsymbol{c}) = \log \sigma(\boldsymbol{f}, \boldsymbol{c}) = \log\left(\frac{1}{1+\exp(-\boldsymbol{f}\cdot\boldsymbol{c})}\right)$. The gradient with respect to $\boldsymbol{f}$ is $\nabla_{\boldsymbol{f}}(L_+(\boldsymbol{f}, \boldsymbol{c})) = \boldsymbol{c}\frac{1}{1+\exp(\boldsymbol{f}\cdot\boldsymbol{c})}$ and the respective update of $\boldsymbol{f}' \leftarrow \boldsymbol{f} + \eta\frac{1}{1+\exp(\boldsymbol{f}\cdot\boldsymbol{c})}\boldsymbol{c}$ clearly increases $\cos_{\text{sim}}(\boldsymbol{f}, \boldsymbol{c})$. Consider two focus entities $1, 2$ and corresponding positive associations with context entity $j$. When positive examples $(1, j)$ and $(2, j)$ are in the same focus-updating minibatch, both $\cos_{\text{sim}}(\boldsymbol{f}_1, \boldsymbol{c})$ and $\cos_{\text{sim}}(\boldsymbol{f}_2, \boldsymbol{c})$ increase and a desirable side effect is that in expectation $\cos_{\text{sim}}(\boldsymbol{f}_1, \boldsymbol{f}_2)$ increases as well. This is achieved when the

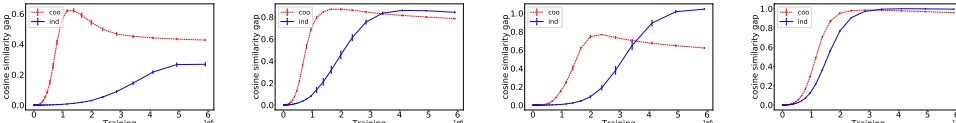

Figure 4: Square $10^4 \times 10^4$ stochastic blocks when training with few example interactions (independent and coordinated samples), with minibatch size $b = 4$. Left to right: $(T = 5, B = 10)$; $(T = 5, B = 100)$, $(T = 20, B = 10)$, $(T = 20, B = 100)$.

updates on corresponding examples $(1, j)$ and $(2, j)$ is performed with the same current parameters $c_j$, which happens with COO (and with full gradients) but less so with IND arrangements that on average place the two examples half an epoch apart. Figure 3 shows the expected increase in cosine similarity $\mathbb{E}\left[\cos_{\text{sim}}(\boldsymbol{f}_1', \boldsymbol{f}_2') - \cos_{\text{sim}}(\boldsymbol{f}_1, \boldsymbol{f}_2)\right]$ for learning rates $\eta = 0.02, 0.05$ when the vectors $\boldsymbol{f}_1$, $\boldsymbol{f}_2$, and $\boldsymbol{c}$ are independently drawn from a product distribution $\mathcal{N}(0, 1)^d$ of independent Gaussians.

**Self similarity** Coordinated arrangements preserve information on entity similarity in fraction of epochs. More formally, consider for two entities the weighted Jaccard similarity computed from examples in a small stretch of training that includes a very small number of examples with each entity. With COO, the expected similarity is equal to that of the full vectors whereas with IND, the similarity information disperses rapidly. This is because COO uses coordinated samples which maximize preserved similarity for the marginal distribution Cohen et al. (2009). It is instructive to consider two focus entities with Jaccard similarity $J$ and $M$ contexts with positive $\kappa_{ij} = c > 0$. An $\alpha \ll 1$ fraction of an epoch will on average include $\alpha M$ sampled contexts from each focus entity. When the samples are independent then the sets would be highly dissimilar even when $J$ is close to 1. When the samples are coordinated then the expected similarity in the sample corresponds to the similarity of the original vectors. We next demonstrate the self similarity quality experimentally with stochastic block matrices. We select small sets of positive training examples using independent and coordinated sampling schemes according to the same per-entity marginal distributions. We then train with this small set on multiple epochs until convergence as a way to gauge the "information" each set provides and its effect on training speed. We sample $T = 5, 10, 15, 20$ example interactions from each row (for focus updates) and symmetrically from each column (for context updates) of the association matrix. With independent sampling we select $T$ independent examples for each row $i$ by selecting a column $j$ with probability $\kappa_{ij}/||\kappa_i.||_1$. For coordinated sampling we repeat the following $T$ times. We draw $u_j \sim \text{Exp}[1]$ for each column and select for each $i$ column $\arg\max_j \kappa_{ij}/u_j$. Clearly the marginal distribution is the same, as the probability that column $j$ is selected for row $i$ is equal to $\kappa_{ij}/||\kappa_i.||_1$. Symmetric schemes apply to columns. We trained embeddings (with IND arrangements) on these smaller sets of examples on otherwise identical setups. Training was one-sided and alternated on each minibatch with row samples used for updating row embeddings and column samples for updating column embeddings. Representative results ($b = 4$) are reported in Figure 4. We observe that the coordinated selection of training examples consistently attains faster convergence in the earlier epochs. With fewer examples per entity, coordinated selection also had a higher peak quality than the respective independent selection. With more examples and larger blocks, the coordinated selection peaked lower, due to loss of the multi-hop expander structure.

# 6 CONCLUSION

We consider embedding computations with stochastic gradients and establish that the arrangement of training examples into minibatches can be a powerful performance knob. In particular, we introduced coordinated arrangements as a principled method to accelerate SGD training of embedding vectors. Our experiments focused on the popular SGNS loss and our methods were designed for pairwise associations. In future we hope to explore the use of coordinated arrangement with other loss objectives, deeper networks, and more complex association structures.

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

## A  ADDITIONAL RESULTS FOR STOCHASTIC BLOCKS

We report here results for minibatch size parameter $b = 4$ and $b = 256$ for the same set of arrangement methods and combinations as reported for $b = 64$ in Section 4.4 (dimension $d = 50$ and block sizes $B = 10, 20, 50, 100$). Convergence with the precision quality measure is shown in Figure 5 ($b = 4$) and Figure 6 ($b = 256$). Convergence for the cosine gap quality measure are shown in Figure 7 ($b = 4$) and Figure 8 ($b = 256$). We observe that the training gain of coordinated arrangements over the baseline IND increases with minibatch size. The methods that cap the microbatch size by the minibatch size (Jaccard* and Angular*) perform much better with larger minibatches, as larger minibatches allow for a higher recall of helpful co-placements. In particular we can see that in early training on small minibatches ($b = 4$) these methods are outperformed by COO (which produces our largest (and unrefined) microbatches).

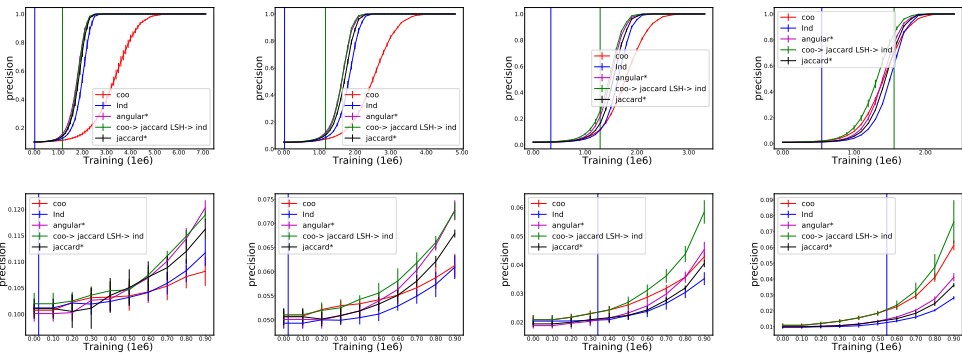

Figure 5: Precision at $k = 10$ with different arrangement methods in the course of training ($d = 50$, $b = 4$). Using $10^4 \times 10^4$ stochastic blocks matrices with $B \in \{10, 20, 50, 100\}$. The switch point for the MIX method are shown in blue (to COO+LSH) and green (to IND). The solid lines are for Jaccard LSH and the dashed lines are for angular LSH.

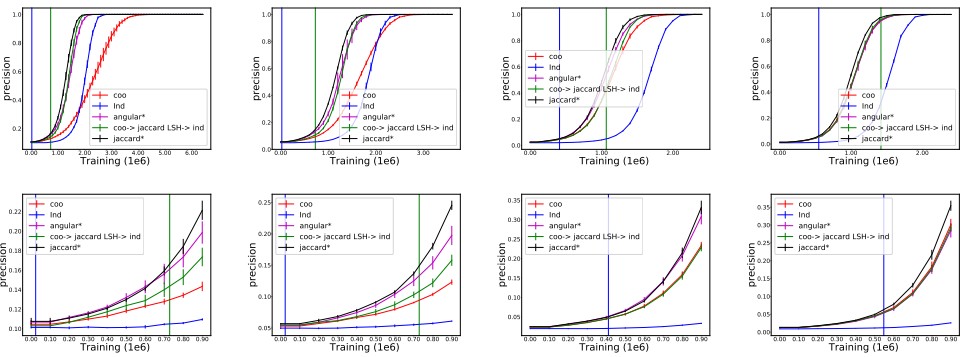

Figure 6: Precision at $k = 10$ with different arrangement methods in the course of training ($d = 50$, $b = 256$). Using $10^4 \times 10^4$ stochastic blocks matrices with $B \in \{10, 20, 50, 100\}$. The switch point for the MIX method are shown in blue (to COO+LSH) and green (to IND). The solid lines are for Jaccard LSH and the dashed lines are for angular LSH.

We quantify the gains of different methods over the baseline IND arrangements in the following tables. Results for precision at $k = 10$ are reported in Table 2 for Jaccard LSH MIX, in Table 3 for (pure) Jaccard*, and in Table 4 for angular* LSH MIX(where angular LSH is applied with respect to a $d = 3$ embedding). Results for cosine gap are reported in Table 5 for Jaccard LSH MIX and in Table 6 for angular* LSH MIX.

The tables report results for different minibatch sizes $b$ and block sizes $B$. In each table we list the peak quality (cosine gap or precision) of the respective coordinated method and the amount of training used by IND to reach 0.75, 0.95, or 0.99 of that peak. We also show the reduction in training

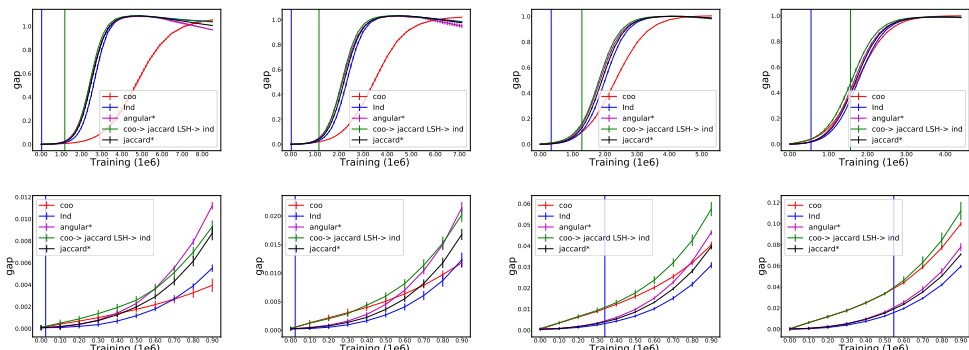

Figure 7: Cosine gap with different arrangement methods in the course of training ($d = 50$, $b = 4$). Using $10^4 \times 10^4$ stochastic blocks matrices with $B \in \{10, 20, 50, 100\}$. The switch point for the MIX method are shown in blue (to COO+LSH) and green (to IND). The solid lines are for Jaccard LSH and the dashed lines are for angular* LSH.

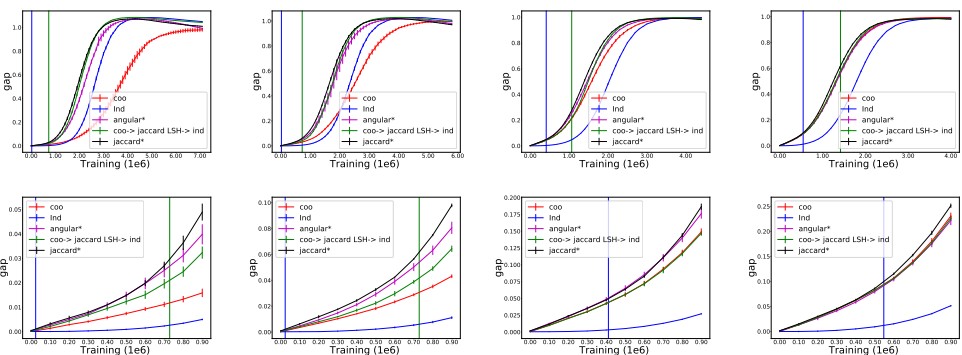

Figure 8: Cosine gap with different arrangement methods in the course of training ($d = 50$, $b = 256$). Using $10^4 \times 10^4$ stochastic blocks matrices with $B \in \{10, 20, 50, 100\}$. The switch point for the MIX method are shown in blue (to COO+LSH) and green (to IND). The solid lines are for Jaccard LSH and the dashed lines are for angular* LSH.

that is gained by using the respective coordinated method instead of IND. Overall, we can see that the coordinated methods consistently had training gains of 5-30%. The gain is larger with smaller blocks and also with larger minibatches. The emphasized numbers in the Jaccard MIX and Jaccard* correspond to the method that provided the higher gain. We can see that Jaccard* performed better than Jaccard MIX for larger minibatch sizes.

## B EMBEDDING DIMENSION ANALYSIS

We report here results on the effect of the dimension on the embedding quality and convergence, focusing on training with IND arrangements. Figure 10 shows quality in the course of training for different dimensions for selected $10^4 \times 10^4$ stochastic block matrices. We show both the cosine gap and the precision with $k = 10$. Figure 9 shows the cosine gap and precision with $k = 10$ quality for the MOVIELENS1M and AMAZON data sets. The precision on the recommendation data sets is computed over focus entities (users) with at least 20 positive entries. The precision is the fraction of top $k$ that are in the test set.

| #blocks $B$ | mbatch size $b$ | peak | 0.75 %gain | $\times 10^6$ | 0.95 %gain | $\times 10^6$ | 0.99 %gain | $\times 10^6$ |
|---|---|---|---|---|---|---|---|---|
| 10 | 4 | 1.00 | 9.63 | 2.18 | 8.61 | 2.44 | **6.61** | 2.57 |
| 10 | 64 | 1.00 | 14.22 | 2.18 | 13.93 | 2.44 | 12.69 | 2.60 |
| 10 | 256 | 1.00 | 28.77 | 2.19 | 26.12 | 2.45 | 25.57 | 2.62 |
| 20 | 4 | 1.00 | 10.00 | 2.00 | 9.87 | 2.23 | **8.94** | 2.35 |
| 20 | 64 | 1.00 | 15.50 | 2.00 | 15.25 | 2.23 | 13.14 | 2.36 |
| 20 | 256 | 1.00 | 29.15 | 1.99 | 26.46 | 2.23 | 24.79 | 2.38 |
| 50 | 4 | 1.00 | 9.50 | 1.79 | 8.04 | 1.99 | **5.77** | 2.08 |
| 50 | 64 | 1.00 | 15.64 | 1.79 | 14.07 | 1.99 | 10.58 | 2.08 |
| 50 | 256 | 1.00 | 28.89 | 1.80 | 26.37 | 2.01 | 23.08 | 2.08 |
| 100 | 4 | 1.00 | 10.30 | 1.65 | 7.65 | 1.83 | **3.66** | 1.91 |
| 100 | 64 | 1.00 | 18.18 | 1.65 | 14.21 | 1.83 | 11.40 | 1.93 |
| 100 | 256 | 1.00 | 28.31 | 1.66 | 24.32 | 1.85 | 21.32 | 1.97 |

Table 2: Training gain of Jaccard MIX arrangement with respect to IND baseline for $10^4 \times 10^4$ stochastic blocks. Peak is maximum *precision at $k = 10$* quality for MIX. We report the training amount for IND to reach 75% , 95%, and 99% of peak with respective percent reduction in training with MIX.

| #blocks $B$ | mbatch size $b$ | peak | 0.75 %gain | $\times 10^6$ | 0.95 %gain | $\times 10^6$ | 0.99 %gain | $\times 10^6$ |
|---|---|---|---|---|---|---|---|---|
| 10 | 4 | 1.00 | 7.34 | 2.18 | 7.00 | 2.43 | 5.77 | 2.60 |
| 10 | 64 | 1.00 | 16.97 | 2.18 | 15.57 | 2.44 | **14.50** | 2.62 |
| 10 | 256 | 1.00 | 33.33 | 2.19 | 30.61 | 2.45 | **29.17** | 2.64 |
| 20 | 4 | 1.00 | 6.03 | 1.99 | 6.31 | 2.22 | 5.04 | 2.38 |
| 20 | 64 | 1.00 | 17.09 | 1.99 | 16.22 | 2.22 | **15.06** | 2.39 |
| 20 | 256 | 1.00 | 35.00 | 2.00 | 31.39 | 2.23 | **29.88** | 2.41 |
| 50 | 4 | 1.00 | 5.59 | 1.79 | 5.53 | 1.99 | 2.84 | 2.11 |
| 50 | 64 | 1.00 | 17.88 | 1.79 | 15.58 | 1.99 | **12.44** | 2.09 |
| 50 | 256 | 1.00 | 35.36 | 1.81 | 30.50 | 2.00 | **26.67** | 2.10 |
| 100 | 4 | 1.00 | 4.24 | 1.65 | 2.73 | 1.83 | 2.58 | 1.94 |
| 100 | 64 | 1.00 | 18.79 | 1.65 | 15.85 | 1.83 | **13.33** | 1.95 |
| 100 | 256 | 1.00 | 31.93 | 1.66 | 26.63 | 1.84 | **24.24** | 1.98 |

Table 3: Training gain of Jaccard* arrangement with respect to IND baseline for $10^4 \times 10^4$ stochastic blocks. Peak is maximum *precision at $k = 10$* quality for Jaccard*. We report the training for IND to reach 75% , 95%, and 99% of peak with respective percent reduction in training with Jaccard*.

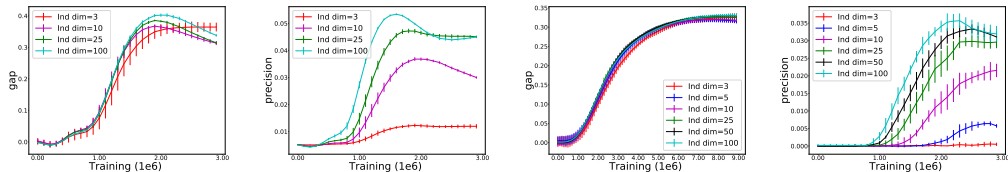

Figure 9: Training (IND with $b = 64$) with different dimensions. From left: MOVIELENS1M (cosine gap and precision for $k = 50$) and AMAZON (cosine gap and precision for $k = 50$).

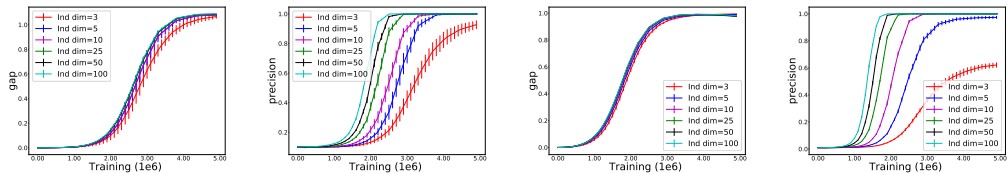

Figure 10: Training (IND $b = 64$) with different dimensions on $10^4 \times 10^4$ Stochastic blocks. From left: $B = 10$ (cosine gap and precision at $k = 10$) and $B = 100$ (cosine gap and precision at $k = 10$).

On all data sets we can observe slightly faster convergence with higher dimension in terms of number of training examples. The per-example training cost, however, increases much faster and proportionally to the dimension. This means that lower dimension is more effective in terms of computation in reaching a particular lower quality level. This supports methods like ours that leverage

| #blocks | mbatch | | 0.75 | | 0.95 | | 0.99 | |
| $B$ | size $b$ | peak | %gain | $\times 10^6$ | %gain | $\times 10^6$ | %gain | $\times 10^6$ |
|---|---|---|---|---|---|---|---|---|
| 10 | 4 | 1.00 | 10.55 | 2.18 | 9.84 | 2.44 | 8.56 | 2.57 |
| 10 | 64 | 1.00 | 11.47 | 2.18 | 10.66 | 2.44 | 11.92 | 2.60 |
| 10 | 256 | 1.00 | 23.29 | 2.19 | 22.45 | 2.45 | 20.99 | 2.62 |
| 20 | 4 | 1.00 | 11.00 | 2.00 | 9.87 | 2.23 | 10.21 | 2.35 |
| 20 | 64 | 1.00 | 12.50 | 2.00 | 11.66 | 2.23 | 12.29 | 2.36 |
| 20 | 256 | 1.00 | 24.62 | 1.99 | 23.32 | 2.23 | 21.43 | 2.38 |
| 50 | 4 | 1.00 | 12.29 | 1.79 | 11.06 | 1.99 | 9.62 | 2.08 |
| 50 | 64 | 1.00 | 16.20 | 1.79 | 14.57 | 1.99 | 11.06 | 2.08 |
| 100 | 4 | 1.00 | 12.12 | 1.65 | 9.84 | 1.83 | 7.33 | 1.91 |
| 100 | 64 | 1.00 | 18.18 | 1.65 | 15.30 | 1.83 | 11.92 | 1.93 |
| 100 | 256 | 1.00 | 28.31 | 1.66 | 24.32 | 1.85 | 22.34 | 1.97 |

Table 4: Training gain of angular* LSH MIX arrangement (based on $d = 3$ embeddings) with respect to IND baseline for $10^4 \times 10^4$ stochastic blocks. Peak is maximum *precision at $k = 10$* quality for MIX. We report the training amount for IND to reach 75% , 95%, and 99% of peak with respective percent reduction in training with MIX.

| #blocks | mbatch | | 0.75 | | 0.95 | | 0.99 | |
| $B$ | size $b$ | peak | %gain | $\times 10^6$ | %gain | $\times 10^6$ | %gain | $\times 10^6$ |
|---|---|---|---|---|---|---|---|---|
| 10 | 4 | 1.09 | 6.58 | 3.04 | 4.90 | 3.67 | 4.76 | 4.20 |
| 10 | 64 | 1.09 | 9.87 | 3.04 | 8.45 | 3.67 | 7.35 | 4.22 |
| 10 | 256 | 1.08 | 20.20 | 3.07 | 17.25 | 3.71 | 16.32 | 4.29 |
| 20 | 64 | 1.03 | 11.36 | 2.73 | 9.28 | 3.34 | 7.77 | 3.86 |
| 20 | 256 | 1.03 | 21.61 | 2.73 | 18.58 | 3.39 | 16.33 | 3.92 |
| 50 | 4 | 1.00 | 8.37 | 2.39 | 6.69 | 2.99 | 5.43 | 3.50 |
| 50 | 64 | 1.00 | 12.97 | 2.39 | 10.00 | 3.00 | 7.43 | 3.50 |
| 50 | 256 | 1.00 | 23.24 | 2.41 | 18.27 | 3.01 | 15.77 | 3.55 |
| 100 | 4 | 0.99 | 8.41 | 2.14 | 6.23 | 2.73 | 5.57 | 3.23 |
| 100 | 64 | 0.99 | 14.02 | 2.14 | 10.58 | 2.74 | 7.74 | 3.23 |
| 100 | 256 | 0.99 | 21.76 | 2.16 | 17.09 | 2.75 | 15.50 | 3.29 |

Table 5: Training gain of Jaccard MIX arrangement with respect to IND arrangement baseline for $10^4 \times 10^4$ stochastic blocks. Peak is maximum *cosine gap* quality for MIX. We report training amount for IND to reach 75% , 95%, and 99% of peak with respective percent reduction in training with MIX.

coarser, lower quality, but much more efficient to compute embeddings to accelerate the training of more complex models.

On the recommendations data sets and for the precision quality measure on the stochastic blocks data we can see that the peak quality increases with the dimension. In particular, we can see that the peak quality for $d = 3$ is considerably lower than for $d = 50$. This means that higher dimension are effective in providing better peak quality. This supports our use in the experiments of the $d = 3$ embedding at the basis of angular* COO+LSH microbatches in order to accelerate the training of $d = 50$ embeddings, which are costlier to train but provide higher peak quality.

| #blocks $B$ | mbatch size $b$ | peak | 0.75 | | 0.95 | | 0.99 | |
|---|---|---|---|---|---|---|---|---|
| | | | %gain | $\times 10^6$ | %gain | $\times 10^6$ | %gain | $\times 10^6$ |
| 10 | 4 | 1.09 | 7.24 | 3.04 | 5.72 | 3.67 | 5.48 | 4.20 |
| 10 | 64 | 1.09 | 8.55 | 3.04 | 6.81 | 3.67 | 6.40 | 4.22 |
| 10 | 256 | 1.08 | 15.31 | 3.07 | 12.40 | 3.71 | 12.59 | 4.29 |
| 20 | 4 | 1.03 | 8.42 | 2.73 | 5.99 | 3.34 | 5.94 | 3.87 |
| 20 | 64 | 1.03 | 9.16 | 2.73 | 7.19 | 3.34 | 6.22 | 3.86 |
| 20 | 256 | 1.03 | 17.95 | 2.73 | 15.04 | 3.39 | 13.01 | 3.92 |
| 50 | 4 | 1.00 | 9.62 | 2.39 | 7.69 | 2.99 | 6.00 | 3.50 |
| 50 | 64 | 1.00 | 12.97 | 2.39 | 9.67 | 3.00 | 7.71 | 3.50 |
| 100 | 4 | 0.99 | 10.28 | 2.14 | 8.06 | 2.73 | 6.81 | 3.23 |
| 100 | 64 | 0.99 | 14.49 | 2.14 | 10.95 | 2.74 | 8.36 | 3.23 |
| 100 | 256 | 0.99 | 22.69 | 2.16 | 17.82 | 2.75 | 15.20 | 3.29 |

Table 6: Training gain of angular* LSH MIX arrangement (based on $d = 3$ embeddings) with respect to IND baseline for $10^4 \times 10^4$ stochastic blocks. Peak is maximum *cosine gap* quality for MIX. We report the amount of training for IND to reach 75% , 95%, and 99% of peak with respective percent reduction in training with MIX.

