# OpenReview forum: "LSH Microbatches for Stochastic Gradients:  Value in Rearrangement"
_ICLR.cc/2019/Conference_

### Official Review · AnonReviewer2 · 2018-11-03
**Experimentally weak with the results not justifying the increased computation. No comparison to other methods doing non-uniform sampling and mini-batch selection.**

**Rating:** 4
**Confidence:** 4

**Review:**

###### Post-Revision ########################
Thank you for revising the paper and addressing the reviewers' concerns. The updated version reads much better and I have updated my score.

Unfortunately, I still think that the experimental analysis is not enough to warrant acceptance. I would encourage the authors to have a more detailed set of experiments to showcase the effectiveness of their method and have ablation studies to disentangle the effects of the different moving parts.
###### Post-Revision ########################

This paper considers arranging the examples into mini-batches so as to accelerate the training of metric embeddings. The
- The paper doesn't have sufficient experimental evidence to convince me that the proposed method is useful. There is no comparison against baselines. The paper is not clearly written or well organized. Detailed comments below:
- For example, when introducing focus and context entities, it would be helpful to give examples of this to make it clearer.
- In section 3, please clarify that after drawing both positive and negative examples, what is the size of the minibatch for which the gradient is calculated?
- How do you choose the size of the microbatches? If the microbatch size is too small, then the effect of associating examples is small.
- In the line, "Instead, we use LSH modules that are available at the start of training and are only a coarse proxy of the target similarity" Why are you not iteratively refining the LSH modules as the training progresses? Won't this lead to an improvement in the performance?
- In the line "The coarse embedding can come from a weaker (and cheaper to train) model or from a partially-trained model. In our experiments we use a lower dimension SGNS model." Could you please clarify what is the additioanal computational complexity of the method? This involves additional computational cost? It doesn't seem to me that the results justify this increased computation. Please justify this.
- In Lemma 3.2, the term s_i is undefined
- "In early training, basic coordinated microbatches with few or no LSH refinements may be most effective. As training progresses we may need to apply more LSH refinements. Eventually, we may hit a regime where IND arrangements dominate." This explanation is vague and has no theoretical or empirical evidence supporting it. Please clarify this.
- Please fix the size of the axes and the legend in all the figures.
- For figure 1, how is the step-size chosen? What is the dimensionality of the examples?
- From figure 3, it is not clear that the proposed methods lead to significant gains over the independently sampling the examples? Are there any savings in the wall clock time for the proposed methods? Why is there no comparison against other methods that have proposed non-uniform sampling of examples for SGD (like Zhang, 2017)? Are the hyper-parameters chosen in a principled way for these experiments?

---

> ### Author Response · Authors · 2018-11-17
> **Response to review**
>
> We submitted a revised manuscript which we hope clarifies issues. It seems that the reviewer missed on some critical aspects of our contribution.    Please see below.
>
> R: "The paper doesn't have sufficient experimental evidence to convince me that the proposed method is useful. There is no comparison against baselines."
> A:  We believe we compared extensively and favourable with the only applicable baseline, which is independent arrangements.  Other methods that the reviewer cites, for example non-uniform sampling such as Zhang 2017,  are orthogonal to our approach and not directly relevant.   Please note that our method is based on rearranging the sampled examples,  but respects a specified marginal distribution over examples.  It can not be compared in a meaningful way to methods that modify the distributions.  Rather, it can be combined with them (this is essentially changing kappa).
>
> R: "The paper is not clearly written or well organized."
> A: We revised extensively to make the paper more accessible to a wider community.
>
> Detailed comments:
>
> R: "when introducing focus and context entities, it would be helpful to give examples..."
> A:  We did in the revision
>
> R: "How do you choose the size of the microbatches? If the microbatch size is too small, then the effect of associating examples is small."
> A: The microbatch size distribution is determined from the generator and the number of LSH maps we apply.  The reviewer is correct there is some recall/precision tradeoff with the number of maps we apply.  We experimented with very basic variants:  No LSH maps (coo), a single map (Jaccard), and applying adaptively until microbatch size is below minibatch size (Jaccard*, Angular*).  Even with these variants we obtained significant improvements over the baseline.
>
> R: "In the line "The coarse embedding can come from a weaker (and cheaper to train) model or from a partially-trained model. In our experiments we use a lower dimension SGNS model." Could you please clarify what is the additioanal computational complexity of the method? This involves additional computational cost? ...  Please justify this."
>
> A:  We showed two use cases for applicable LSH map.  We obtained improvements with both.  One based on the Jaccard similarity of association vectors and one that can come from a weaker model.  In our experiments we used a 3 dimensional embedding (the computation is much lower than computing a 50 dimensional embedding).   This is more of a proof of concepts.   With both methods we obtained significant reduction in training with respect to independent arrangements baseline.    The revised version also includes some analysis of the computational cost of preprocessing that is needed to efficiently generate our microbatches.   The computation is linear in the sparsity of kapps and is negligible compared to training computation.
>
> R:  "In Lemma 3.2, the term s_i is undefined"
> A: The notation s_i (key according to LSH map) was defined earlier, but in our revision it is reviewed just before the statement of the Lemma.
>
> R: ""In early training, basic coordinated microbatches with few or no LSH refinements may be most effective. As training progresses we may need to apply more LSH refinements. Eventually, we may hit a regime where IND arrangements dominate." This explanation is vague and has no theoretical or empirical evidence supporting it. Please clarify this."
>
> A:  Thank you.  For thoutoughly revised this section and this sentence.  Clarification:  By applying LSH refinements we increase the "quality" of the microbatch but we decrease its recall.  The point is that the balance is different in different regimes of the training.   Early on we want more recall and later we want more precision.
>
>
> R: "From figure 3, it is not clear that the proposed methods lead to significant gains over the independently sampling the examples? Are there any savings in the wall clock time for the proposed methods?"
>
> A: Our experiments show that gain of the proposed methods varies according to block size (stochastic blocks) and  mini batch size (increases with minibatch size).  The gains (sometimes over 20%) are fairly significant and also interesting conceptualy.  The revision includes a discussion of the computation overhead of supporting the generation of our minibatches (we show it is linear in the sparsity of kappa).  That overhead is insignificant compared with training, so essentially the training gains should translate to wall clock time gains.
>
> R: "Are the hyper-parameters chosen in a principled way for these experiments?"
> A: Hyperparameters such as learning rate were tuned to work well on the baseline independent arrangements and then we used the same values with all methods.  Our "mix" method had additional hyperparameters which is the switch points  (start with coo, then LSH map, then IND).  Those were determined on one split (or generated model) and applied to others.

---

### Official Review · AnonReviewer4 · 2018-11-14
**Lacks comprehensive results on real-world data sets, writing does not seem revised**

**Rating:** 3
**Confidence:** 2

**Review:**

This paper proposes using structured mini-batches to speed up learning of embeddings. However, it lacks sufficient context with previous work and bench-marking to prove the speed-up. Furthermore, it is difficult to read due to its lack of revision. Sentences are wordy and do not always have sufficient detail.

Argument issues and questions:
- Since the main claim is a speed-up in training, the authors should support with robust experimentation. Only a synthetic and small test are conducted.
- Not being an expert in this subject, it was difficult to follow some of the ideas of the paper. They were presented without clear explanation of why they supported the conclusion. For example,  I do not understand Figure 4. It seems COO and IND change places. It is not always clear how the figures support the argument.
- What impact does the size of the micro-batch have on the speed-up?
- How does this approach compare to other embedding approaches in terms of speed? There are no benchmarks other than IND.

Formatting issues:
- On page one, the sentence "We make a novel case here for the antithesis of coordinated arrangements,
where corresponding associations are much more likely to be included in the same minibatch" seems contradictory. It reads that you are arguing for "the antithesis of coordinated arrangements, namely independent arrangements" when you mean "the antithesis of independent arrangements, namely coordinated arrangements."
- The figures in this paper are all very small with minuscule text and legends. Only after zooming in 200% were they legible. Figure 3, 4, 5, 6, and 7 have no axis labels. It is sometimes clear from the caption what the axes are, but it is hard to follow.
- Often references are cited in the text without being set off with parentheses or grammatical support. For example at the top of page three: "One-sided updates were introduced with alternating minimization Csiszar & Tusnády (1984) and for our purposes they facilitate coordinated arrangements and allow more precise matching of corresponding sets of negative examples to positive ones." This interrupts the sentence making it hard to read.

---

> ### Author Response · Authors · 2018-11-17
> **Response to reviewer concerns**
>
> R: "This paper proposes using structured mini-batches to speed up learning of embeddings. However, it lacks sufficient context with previous work and bench-marking to prove the speed-up."
>
> A: As far as we know, our approach (rearranging the minibatches, same marginal distribution) is novel and there isn't really relevant previous benchmarks except for the baseline of independent at random placement.  We do compare very favourably with this baseline.  Most other optimizations we are aware of that may seem related are orthogonal and generally  unhelpful on our "symmetric" synthetic data.
>
> R: "Furthermore, it is difficult to read due to its lack of revision. Sentences are wordy and do not always have sufficient detail."
> A:  Our approach draws on concept from sketching and sampling theory that may not be familiar to many ICLR reviewers.  We uploaded a significant revision that we hope makes the paper more accessible.
>
> R: "Since the main claim is a speed-up in training, the authors should support with robust experimentation. Only a synthetic and small test are conducted."
> A:  This is the first paper on a novel approach and meant to be a proof of concept.  We believe the results on the synthetic and recommendation data sets are very meaningful.  The speedup is evident because it is proportional to the amount of training, which is reduced.
>
> R: Not being an expert in this subject, it was difficult to follow some of the ideas of the paper. They were presented without clear explanation of why they supported the conclusion. For example,  I do not understand Figure 4. It seems COO and IND change places. It is not always clear how the figures support the argument.
> A:  We revised the writing to make it more accessible.  Yes, COO and IND sometimes change places because they have properties that balance out differently at different regimes of the training.  We revised so this is explained better now.   When we add LSH maps to COO we obtain methods that always dominate the baseline IND.
>
> R: What impact does the size of the micro-batch have on the speed-up?
> A: The revised version includes results for different minibatch sizes and we obtain improvements for all. The benefit of our methods with respect to the baseline IND increases with minibatch size.   The size of the microbatch can impact performance in some regimes when it is larger than the minibatch size.  In this case the LSH refinements address this.
>
> R: How does this approach compare to other embedding approaches in terms of speed? There are no benchmarks other than IND.
> A:  We believe IND is the only relevant benchmark.  There are proposed approaches that alter the marginal distribution of examples, but they are orthogonal to our approach (can be combined with it).   We work with the same marginal distribution and only alter the arrangement.  We are not aware of other baselines that only alter arrangements.
>
> R: Formatting issues
> A: mostly addressed in the revision.  Thank you!

---

### Official Review · AnonReviewer3 · 2018-11-15
**limited novelty and experimental results**

**Rating:** 4
**Confidence:** 4

**Review:**

This paper discussed a non-uniform sampling strategy to construct minibatches in SGD for the task of learning embeddings for object associations. An example throughout the paper is learning embeddings for a set F of focus entities and set C of context entities. In general, for focus update, the algorithm draws for each minibatch certain amount of positive samples (i,j), i \in F and j \in C. Then for each positive pair, we select certain amount of negative samples (i,j’) for j’ \in some uniformly randomly selected subset C’. The same algorithm is implemented for context update, and the training alternates between the two. The authors choose similar positive object in one minibatch since it’s more efficient. Therefore, LSH hashing is used to point similar items to similar keys. Two similarity measures are used here, Jaccard similarity and cosine similarity. Some experiments are demonstrated on synthetic data and two real datasets to show the effectiveness of their method.

Concerns:
1.	Every piece of the method has been well studied, and the combination of them proposed in this paper does not seem very novel.
2.	Algorithm 4, which is the hashing for Jaccard similarity, seems wrong. Only using iid exponentials cannot make collision probability equal Jaccard similarity.
3.	Little experiments on real datasets. No comparison with other non-uniform minibatch construction methods (there should be some).
4.	No quantitative analysis.
5.	Structure of the paper could be improved. For example, it’s better to put section 4 and 6 together.

---

> ### Author Response · Authors · 2018-11-15
> **Response to reviewer concerns**
>
> We disagree with the reviewer determination of "limited novelty".  This is the first time that the arrangement of examples is considered for SGD in a principled way.  It is orthogonal to and very different than other optimizations. We also believe that our experimental results, because of the simplicity and properties of the synthetic data,  are particularly meaningful and demonstrate well the strength and potential of the approach.
>
> 1.
>   R:  "every piece of the method had been well studied".
>   A: Our claimed contribution here is the concept of arranging training examples in a principled way. We explain why it is beneficial, provide efficient methods of doing so, and experimentally demonstrate the benefit.  This notion of "arrangement methods", as far as we know, is novel.  Yes, the general setting we consider and our experiments are on top of a well studied method (SGNS model,  "gravity" like negative sampling,....)
>
>  2.
>  R: "Algorithm 4 seems wrong."
>  E: We disagree.  You can follow the references and proofs if you wish. To obtain intuition why, note that for focus updates (symmetrically for context) the iid exponentials are drawn per context entity to essentially select a sample of contexts for each focus.  The coordination is achieved by using the same assignment of randomization to contexts across focus entities. At the extreme, it is easy to verify that when two focus entities have exactly the same set of weighted associations (Jaccard similarity 1), they would have the same sampled context (the one with minimum ratio of iid exponential and weight).  If you would like we can provide further details.
>
>  3.
> R: "No comparison with other non-uniform construction methods"
> A: Can you please be specific?  Please note that our method is orthogonal to prevalent  approaches that alter the marginal distribution of training examples (we cite some of these methods).   We only modify the *arrangement* (the marginal distribution stays the same!).   Moreover,  our use of stochastic blocks synthetic data (the role of all parameters and embedded entities is symmetric) factors out the benefit of the bulk of other optimizations (such as per-parameter learning rate and re-weighting of examples).  We are not aware of literature on other methods of altering the arrangement that we could compare with other than the baseline of "independent" groupings.
>
> R: Little experiments on real datasets
> A: Yes, we included some, and hope to do more in future. At this preliminary stage of introducing a new approach, we found synthetic datasets to be valuable in that they provide a clean well understood test bed.
>
>  4.
>  R: "No quantitative analysis"
>  A: Could you please elaborate what you are looking for?  (We do establish correctness and properties of our arrangement methods.  We now also include a more elaborate analysis of the computation involved).
>
>  5.
>  R: "Structure of the paper could be improved"
>  We uploaded a modified version and will be happy to address any further concerns.

---

### Official Review · AnonReviewer5 · 2018-11-15
**Requires further clarification and empirical justification**

**Rating:** 4
**Confidence:** 3

**Review:**

The paper presents a method for improving the convergence rate of Stochastic Gradient Descent for learning embeddings by grouping similar training samples together. The basic idea is that gradients computed on a batch of highly associated samples encode related information in a single update that independent samples might take multiple updates to capture. These structured minibatches are constructed by independently combining subsets of positive examples called “microbatches”. Two methods are presented for constructing these microbaches; first by grouping positive examples by shared context (called “basic” microbatches), second by applying Locality Sensitive Hashing to further partition the microbatches into groups that are more likely to contain similar examples.

Three datasets are used for experimental analysis: a synthetic dataset generated using the stochastic block model, and two large scale recommendation datasets. The presented algorithms are compared to a baseline of independently sampled minibatches using the cosine gap and precision for the top k predictions. The authors show the measured cosine gaps over the course of training as well as the gains in training performance for several sets of hyperparameters.

The motivation and basic intuition behind the work is clearly presented in the introductory section. The theoretical justification for the structured minibatches is reasonably convincing and invites empirical verification.

General concerns:
Any method for improving the performance of an optimization process via additional preprocessing must show that the additional overhead incurred from preprocessing the data (in this case, organizing the minibatches) does not negate the achieved improvement in convergence time. This work presents no evidence that this is the case. I expected to see 1) time complexity analysis of each new algorithm proposed for preprocessing and 2) experimental results showing that the overall computation time, including the proposed preprocessing steps, was reduced by this method. Neither of these things are present in this work.

Furthermore, the measured “training gains” are, to my knowledge, not clearly defined. I assume that the authors are using the number of epochs or iterations before convergence as their measure of training performance, but this should be stated explicitly rather than implicitly.

Finally, the experimental results presented do not seem to entirely support the authors’ conclusions. Figures 2, 3, and 4, as well as several of the figures in the appendix, show some parameter settings for which the gains over the baseline are quite limited. This makes me suspect that perhaps the coordinated minibatches aren’t the only variable affecting performance.

I have organized my remaining minor concerns and requests for clarification by section, detailed below.

Section 1
- In the last paragraph, the acronym SGNS is mentioned before being defined. You should either state the full name of the method (with citation) or omit the mention altogether.

Section 2
- I would like a few sentences of additional clarification on what “focus” entities vs. “context” entities are in the more general case. I am familiar with what they mean in the context of Skip Gram, but I think more discussion on how this generalizes is necessary here. Same goes for what kappa (“association strength”) means, especially considering that this concept isn’t really present (to my understanding) in Skip Gram.
- Grammar correction:
“The negative examples provide an antigravity effect that prevents all embeddings to collapse into the same vector”
“to collapse” -> “from collapsing”

Section 3
- Maybe this is just me, but I find the mu-beta notation for the microbatch distributions rather odd. Why not just use a single symbol?
- I would like a bit more clarification on the proof for lemma 3.1, specifically on the last sentence, “the product of these events …”; that statement did not follow obviously to me.

Section 3.1
- Remove the period and colon near kappas at the end of paragraph 3. It’s visually confusing with the dot index notation right next to them.

Section 4
- Typo: “We selects a row vector …” -> “We select a row vector …”

Section 5
- I don’t understand what Figure 1 is trying to demonstrate. It doesn’t do anything (as far as I can tell) to defend the authors’ claim that COO provides a higher expected increase in cosine similarity than IND.

Section 6
- All figures in this section should have axis labels. The captions don’t sufficiently explain what they are.

Section 6.2
- How is kappa computed for the recommendations datasets? This isn’t obvious at all.

---

> ### Author Response · Authors · 2018-11-15
> **Response addressing the reviewer concerns (see revision also)**
>
>   Thank you so much for the constructive comments. We revised accordingly and hope the concerns are addressed.  This is a novel and potentially powerful optimization to SGD that is orthogonal to other optimizations.  Here is our response.
>
> R: "Any method for improving the performance of an optimization process via additional preprocessing must show that the additional overhead incurred from preprocessing the data (in this case, organizing the minibatches) does not negate the achieved improvement in convergence time. This work presents no evidence that this is the case. I expected to see 1) time complexity analysis of each new algorithm proposed for preprocessing and 2) experimental results showing that the overall computation time, including the proposed preprocessing steps, was reduced by this method. Neither of these things are present in this work."
>
> A: This is an excellent point which we now convey better.  We revised the manuscript to include more details on the computational cost of the preprocessing we require in order to efficiently generate microbatches. To summarize:  The preprocessing computation is linear in the sparsity of $\kappa$ and after the preprocessing and indexing, the generator is very efficient.  What essentially is done is to sort columns/rows by value for basic Microbatch generation and also precompute a small number of LSH maps. Because most applications of embeddings anyhow aggregate and preprocess the raw data (provided associations), for example, by reweighing according to frequency this additional overhead is fairly minimal addition to this preprocessing. Finally, anyhow most of the computation cost tends to occur in the training phase rather than in the preprocessing.  This should be the same per epoch because our generation of Microbatches given the preprocessing is very efficient.   So the gains in training should correspond pretty closely to gains in wall clock time.
> In our work we used a proof-of-concept Python code for microbatch generation that is not suitable for large scale implementation so it was not meaningful to provide wall clock running times.
>
> That said, I must add that (based on decades of experience) I strongly disagree with the reviewer's approach that such insights are only worthwhile if bundled with demonstrated low overhead (which we anyhow did in this case).  This limited thinking is correct when launching a product, but here I hope we are doing research, and novel insights have a stand-alone value that may lead to future work with scalable approaches.
>
> R: "Furthermore, the measured “training gains” are, to my knowledge, not clearly defined. I assume that the authors are using the number of epochs or iterations before convergence as their measure of training performance, but this should be stated explicitly rather than implicitly."
>
> A: Yes, the x-axis shows the total number (with multiplicities) of processed positive examples.  This is proportional to epochs.  The gain is with respect to that.  Thank you for pointing out this was not clear, we added this clarification.
>
> R: "Finally, the experimental results presented do not seem to entirely support the authors’ conclusions. Figures 2, 3, and 4, as well as several of the figures in the appendix, show some parameter settings for which the gains over the baseline are quite limited. This makes me suspect that perhaps the coordinated minibatches aren’t the only variable affecting performance"
>
>  First, Figure 2 shows a different experiment intended to demonstrates the information content in fraction of epochs. So we discuss Figures 3 and 4 that show performance of our main methods and the baseline "ind" for two different metrics.  Please note that we did not select what to show.  We generated stochastic block matrices with a range of block sizes / number of blocks.  Any hyperparameters were either fixed (minibatch sizes, embedding dimension) or tuned to achieve best performance on the *baseline* IND method (learning rate).   Then we simply run our methods and show and discuss the results.  The only additional hyperparameters were with our methods that used "mixed" arrangements.  But the pure (Jaccard*) method was the best performer.  We made this much clearer in the revision and significantly improved the discussion of the experimental results.
>
>  Minor concerns:   All (sections 1-4) were addressed in the revision (will do the figures in Section 6).
>
> As for questions:
> Figure 1 in Section 5 simply demonstrated what we refer to as a "micro level" benefit of coordination, which might be obvious: That even with random assignment, gradient updates involving two entities towards the same (random) entity brings them closer. (In contrast, if the two updates are half an epoch apart then the context embedding is meanwhile modified, so this weakens this effects).  The revision made this clearer.
>
> Section 6.2: The computation of kappa for the recommendations datasets is explained now.

---

### Author Response · Authors · 2018-11-23
**See revised version**

We are thankful for all constructive comments.  We uploaded a revised version with overall  improvements to the presentation and addressing the reviewers' points and suggestions.    We also (i) reorganized the paper where the section that provides principled discussion of the benefit of our methods is now provided after the experiments (ii) included additional results in the appendix (iii) included further explanations to address misunderstandings.
We hope to continue the discussion.

 Some reviewers were concerned with comparison to other approaches. Our paper proposes and demonstrated the potential of a novel approach to SGD optimization.   We are referencing some approaches that are orthogonal, but there is no other baseline for experimental comparison that we are aware of other than independent arrangements (see individual responses).

 Some reviewers were concerned with the overhead of our method.  This is a relevant point that we did not convey well enough in the submission, our revision includes a more thorough analysis.  Overall, the overhead is small, and certainly justifies the improvements.

 Reviewer4 expressed concerns on novelty and correctness of Algorithm 4.  The algorithm is correct, we have references to the works we are relying on,  we included some intuitive discussion in our response that might help the reviewer understanding, and will be happy to provide more.  As for novelty,   we ask the reviewer to point out references supporting their claim. We are not aware of any.

---

### Meta-Review · Area_Chair1 · 2018-12-15
**Not ready for publication ICLR**

**Confidence:** 5
**Recommendation:** Reject

**Metareview:**

Following the unanimous vote of the four submitted reviews, this paper is not ready for publication at ICLR. Among other concerns raised, the experiments need significant work.